# Mental Disorder Symptoms and the Relationship with Resilience among Paramedics in a Single Canadian Site

**DOI:** 10.3390/ijerph19084879

**Published:** 2022-04-17

**Authors:** Justin Mausz, Elizabeth Anne Donnelly, Sandra Moll, Sheila Harms, Meghan McConnell

**Affiliations:** 1Peel Regional Paramedic Services, Operations, Fernforest Division, 1600 Bovaird Drive East, Brampton, ON L6V 4R5, Canada; 2Department of Health Research Methods, Evidence, and Impact, McMaster University, 1280 Main Street West, HSC-2C1, Hamilton, ON L8S 4K1, Canada; 3School of Social Work, The University of Windsor, 167 Ferry Street, Room 167, Windsor, ON N9A 0C5, Canada; donnelly@uwindsor.ca; 4School of Rehabilitation Sciences, McMaster University, 1400 Main Street West, Institute for Applied Health Sciences (IAHS) Building, Room 403, Hamilton, ON L8S 1C7, Canada; molls@mcmaster.ca; 5Department of Psychiatry and Behavioural Neurosciences, McMaster University, 100 West 5th Street, Hamilton, ON L8N 3K7, Canada; harmssh@hhsc.ca; 6Department of Innovation in Medical Education, Faculty of Medicine, The University of Ottawa, 850 Peter Morand Crescent, Room 102, Ottawa, ON K1G 5Z3, Canada; meghan.mcconnell@uottawa.ca

**Keywords:** public safety personnel, first responders, mental disorders, mental health, wellbeing, trauma, operational stress injuries, post-traumatic stress injuries, resilience, peer support

## Abstract

There is growing recognition in research and policy of a mental health crisis among Canada’s paramedics; however, despite this, epidemiological surveillance of the problem is in its infancy. Just weeks before the emergence of the COVID-19 pandemic, we surveyed paramedics from a single, large, urban paramedic service in Ontario, Canada to assess for symptom clusters consistent with post-traumatic stress disorder (PTSD), major depressive disorder, and generalized anxiety disorder and to identify potential risk factors for each. In total, we received 589 completed surveys (97% completion rate) and found that 11% screened positive for PTSD, 15% screened positive for major depressive disorder, and 15% screened positive for generalized anxiety disorder, with one in four active-duty paramedics screening positive for any of the three as recently as February 2020. In adjusted analyses, the risk of a positive screen varied as a function of employment classification, gender, self-reported resilience, and previous experience as a member of the service’s peer support team. Our findings support the position that paramedics screen positive for mental disorders at high rates—a problem likely to have worsened since the onset of the COVID-19 pandemic. We echo the calls of researchers and policymakers for urgent action to support paramedic mental health in Canada.

## 1. Introduction

A 2012 systematic review and meta-analysis calculated the international pooled prevalence of post-traumatic stress disorder (PTSD) among “rescue workers” at 10% [1]. When stratified by occupation, “ambulance workers” were found to have a pooled prevalence of PTSD of 14% [1]. In fact, the “ambulance worker” stratum was used as the reference category in the authors’ relative risk modeling, with Berger et al. suggesting that “ambulance personnel” may be “more susceptible to PTSD” (p. 1009) [1]. The findings parallel a growing body of research internationally that points to high rates of *mental disorders* among emergency medical services (EMS) workers [2,3,4,5,6,7,8], both generally and comparatively within *first responder* occupations. Addressing a conspicuous absence of similar research in Canada, a recent national survey of *public safety personnel* (PSP) found that one in four participating paramedics screened positive for PTSD, one in three screened positive for major depressive disorder, and one in three screened positive for an anxiety disorder, with nearly half screening positive for any one mental disorder [9] and more frequent positive screens observed among women. Related research has also suggested that exposure to *potentially psychologically traumatic events* [10,11,12,13,14,15,16], symptoms consistent with chronic pain [17], clinical insomnia [18], and alcohol use disorder [9], suicidal ideation, planning, and attempts [19], and a history of adverse childhood experiences [20] are prevalent among paramedics, again varying across demographic categories, including gender. The downstream consequences are potentially significant, and can include lost time from work [21], family hardship [22], reduced quality of life [23,24,25,26], and suicidality [7,27], all of which can contribute to burnout [6,28], workplace incivility [29,30], and attrition [31], as well as potentially compromise patient care [32]. As a result, the situation has been characterized in research and policy as a “crisis in Canada” [33,34].

One response to the problem has been a growing interest in workplace resiliency training programs. Resilience is generally held to be the ability for individuals to “bounce back” from adversity [35], and the resilience narrative suggests that (1) individuals can cultivate skills that enhance their resilience, and (2) “more resilient” individuals may be less susceptible to mental health challenges, owing to proactive psychological protection [36,37,38,39,40]. The result has been an increasing adoption of workplace resiliency training within the public safety professions through programs such as the Road to Mental Readiness (R2MR) developed by the Canadian Armed Forces [41], its civilian analogue The Working Mind [42], and a new, PSP-specific program called Before Operational Stress [40]. Evidence supporting the resilience hypothesis, however, has generally been underwhelming, with recent research showing only small effects on mental health literacy, stigma reduction, or mental disorder symptoms [43,44,45].

Any response to the mental health crisis within the paramedic community requires a nuanced understanding of the epidemiology mental disorders among paramedics. Although the extant research in Canada paints a concerning picture, there are methodological limitations that cloud its interpretation. In generating the dataset, the research team cast an intentionally broad net, using a combination of social media advertising and employer and paramedic association list servers to recruit participants. In total, approximately 600 paramedics from across Canada participated, but both the response rate and the characteristics of the population from which the sample was drawn are unknown. This makes understanding the scope of the problem challenging, as results from self-selected samples can be difficult to interpret. It may be, for example, that paramedics with (current or former) mental health challenges are more likely to volunteer for survey research on the topic. Conversely, the opposite may be true. In either case, however, the precision of the existing prevalence estimates—and, by extension, our understanding of risk factors—may be called into question when the representativeness of study samples is uncertain.

### Objectives

Therefore, our objective was to estimate the proportion of symptom clusters consistent with various mental disorders among paramedics in a single, large, urban paramedic service in Ontario, Canada. Our study draws on and (to a degree) replicates the work of Carleton and colleagues [9] on a smaller scale while carefully controlling participant selection within a single site. We hypothesized that, in the context of carefully controlled participant selection, the prevalence of mental disorder symptom clusters would be lower than previously reported in this population in Canada. Lastly, given the growing interest in the resilience narrative in the population, we also sought to explore the relationship between self-reported resilience on the risk of a positive screen for a mental disorder.

## 2. Methods

### 2.1. Overview

Our study took place prior to COVID-19 between September 2019 and February 2020. We distributed an in-person cross-sectional survey to a single, large, urban paramedic service in Ontario, Canada. In addition to a demographic questionnaire, our survey contained validated self-report symptom measures for PTSD, generalized anxiety disorder, and major depressive disorder, as well as a self-report measure for resilience. Our study received ethics approval from the Hamilton Integrated Research Ethics Board (HiREB protocol number 7595), and all respondents provided informed consent to participate. Italicized terms are in reference to the definitions provided by the Canadian Institute for Public Safety Research and Treatment in their Glossary of Terms (version 2.1) [46].

### 2.2. Setting and Participants

We conducted our study in Peel Region, Ontario, Canada. Peel Regional Paramedic Services is the publicly funded sole provider of land ambulance and paramedic services to the municipalities of Brampton, Mississauga, and Caledon, with a total population of 1.3 million residents across a mixed suburban and rural geography of 1200 km^2^. At the time of recruitment, the service employed a total of 714 paramedics responding to approximately 130,000 emergency calls annually, making the service the second largest in the province by staffing and caseload. Workplace resiliency training in the form of the R2MR program [41] was launched in the service in 2017.

Paramedics in Peel Region are required to complete semiannual continuing medical education (CME) days. We distributed our survey during the fall 2019/winter 2020 CME sessions. Following a brief presentation by the principal investigator, surveys were distributed to every paramedic, and consenting paramedics were given approximately 20 min at the beginning of the day to complete the (paper) survey, filling in their responses with black ink pen. The allotted time was selected following a pilot testing phase with a convenience sample of 10 paramedics (not included in the analysis) from different paramedic services to ensure that the participants would have sufficient time to complete the survey. All participants were assured of confidentiality (but not necessarily strict anonymity), and they were given a list of mental health resources available in the community and a 10 CAD Tim Horton’s gift card. Completed surveys were sealed in an opaque envelope and deposited into a locked study drop box. Attendees who did not want to complete the survey were instructed to drop their blank survey package in the same locked drop box as completed surveys but were free to keep the gift card with our thanks. This recruitment strategy was used previously among paramedic services in Ontario [47] and was specifically chosen for its potential to generate high response rates.

### 2.3. Survey

Unlike in previous studies, in which participants could complete surveys over multiple sittings, our participants had only a limited amount of time during their CME to participate. This necessarily constrained the number, length, and complexity of questionnaires we could include in our survey. Each instrument is described below.

#### 2.3.1. Demographics

Our demographic questionnaire was developed through consensus among the research team and was intended to gather data on criteria identified in the literature to be associated with *mental disorder symptoms* among paramedics. This included age, gender, relationship status, education, provider classification (primary or advanced care), employment classification (full vs. part-time), years of experience, and current or previous participation in the service’s peer support team. The peer support team was established in 2017 and was originally intended to provide nonclinical, empathetic support for home and family stressors.

We decided not to collect data on race or ethnicity. While the link between race and ethnicity and health is well established [48,49], and other studies of symptom clusters consistent with mental disorders among paramedics have included ethnicity as a potential predictor [9], the relative racial and ethnic homogeneity of our sample (an extrapolation based on similar research [9,50,51,52]) would likely have left our study underpowered to detect such an effect.

#### 2.3.2. Resilience

We evaluated self-reported resilience using the five-item Brief Resilience Scale (BRS) [53]. The BRS asks respondents to rate their agreement with various statements that characterize how well they recover from adversity (i.e., “I tend to bounce back quickly after hard times”). Response options ranged from 1 (“strongly disagree”) to 5 (“strongly agree”). After reverse-coding three items, the scores were summed and divided by the number of items answered. The scale categorizes respondents into “low” (<3), “normal” (3.00–4.30), or “high” (>4.31) levels of resilience [53] and has been used in at least one previous study among *Public Safety Personnel* in Canada [45]. On the basis of the resilience narrative, we hypothesized that self-reported resilience scores would be inversely associated with the risk of a positive screen for a *mental disorder*.

#### 2.3.3. Self-Report Symptom Measures

The PTSD Checklist-5 (PCL-5; [54]) is a 20-item self-report measure that assesses four criteria specified in the Diagnostic and Statistical Manual (DSM) version 5 for diagnosing PTSD: intrusion, avoidance, alterations in cognition or mood, and arousal or reactivity. Typically, the diagnosis of (or screening for) PTSD is made in reference to an index trauma (“Criterion A”), evaluated using a separate questionnaire. However, given the increasing recognition in scholarship and policy that public safety personnel encounter multiple *potentially psychologically traumatic events* during their work [10,13], we omitted the Criterion A screen. Participants were instead told that the questionnaire asked about “problems that first responders sometimes have in response to a stressful work experience” and asked to rate the frequency with which they had been bothered by symptoms in relation to the stressful work experience in the past 30 days. Symptoms were rated on a five-point anchored scale from 0 (“not at all”) to 5 (“extremely”). Possible scores ranged from 0 to 80, with a summed score >31–33 providing a sensitivity of 88% and a specificity of 69% for probable PTSD [55] when compared to clinical interviews. Consistent with recommendations from the National Center for PTSD [56], we used a cutoff score of >32 to indicate a positive screen for PTSD.

To evaluate for symptoms of major depressive disorder, we used the nine-item Patient Health Questionnaire (PHQ-9) [57]. The PHQ-9 assesses the degree to which depressive symptoms (such as loss of interest or difficulty concentrating) have affected the respondent over the past 14 days. Symptoms are rated on a four-point anchored scale from 0 (“not at all”) to 3 (“nearly every day”), with summed scores >9 corresponding to 85% sensitivity and 82% specificity for probable depression when compared to clinical interviews [58].

Lastly, to evaluate symptoms of generalized anxiety disorder, we used the seven-item Generalized Anxiety Disorder (GAD-7) scale [59]. The GAD-7 assesses the degree to which symptoms of anxiety (such as feeling on edge) have affected the respondent over the last 14 days on a four-point anchored scale from 0 (“not at all”) to 3 (“nearly every day”) with summed scores >9 corresponding to 89% sensitivity and 82% specificity for probable generalized anxiety disorder when compared to clinical interviews [59].

### 2.4. Analyses

We used descriptive statistics to characterize our data, including measures of central tendency, distribution (e.g., skewness and kurtosis), and dispersion for continuous variables and counts and percentages for categorical data. To explore group differences among our participants, we used one-way analysis of variance (ANOVA) and chi-square tests for continuous and categorical data, respectively. To evaluate the internal consistency of the self-report measures, we calculated Cronbach’s alpha for each screening tool.

We used logistic regression modeling to explore the relationship between demographic characteristics and self-reported resilience (collectively, independent variables) on the risk of positive *mental disorder* screens (our dependent variables), including a composite outcome of any positive mental disorder screen. We first constructed unadjusted, univariate logistic regression models to test the association between each demographic variable and our outcomes of interest. Given the exploratory nature of our study, we then entered all demographic variables into adjusted logistic regression models, making no effort to organize the variables hierarchically on the basis of theoretical or statistical significance. Our goal was to assess the specific contributions of the variables included in the study where everything was held constant. Where we made an effort at organizing our model parameters was in the exploration of interaction effects. Here, we constructed interaction terms on the basis of group differences in demographic variables and their association with our outcomes of interest in our unadjusted models. Importantly, our primary interest was in the individual odds ratios of the covariates we included, rather than the predictive capacity of the models as a whole.

## 3. Results

### 3.1. Participation and Response Rate

We distributed a total of 607 surveys to paramedics attending the fall 2019 CME sessions, of which 600 completed surveys were returned. Of these, we excluded 11 for large portions of incomplete data, leaving a final sample of 589 surveys for analysis, corresponding to a response rate of 98.8% and a completion rate of 97%.

During the CME sessions, a total of 107 paramedics (15% of the total workforce) were on long-term leave (Figure 1). Although we originally intended to distribute surveys to paramedics on leave via postal mail early in the new year, personnel within the service who could have facilitated this stage of recruitment were redeployed in response to the COVID-19 pandemic, and we were unable to contact paramedics on leave.

### 3.2. Participant Characteristics

In total, 354 of our participants (60.1%) were men, 232 (39.3%) were women, and a small number (not reported to preserve anonymity) provided another, nonbinary gender. The participants were on average 34.58 (±8.21) years of age and reported an average of 9.30 (±0.44) years of experience as paramedics. The majority (59.1%) of our participants were married or living common-law (later collapsed into single (143; 24.3%) or relationship (446; 75.7%)), had completed a college diploma as their highest education (49.2%), and were working full-time (66.8%), in a front-line role (93.8%), and practicing at the primary care paramedic certification (67.7%). Five percent (*n* = 29) of our participants reported being a current or former member of the service’s peer support team.

When stratified by gender, women were on average younger (33.61 vs. 35.13 years of age, *F* = 5.35; *p* = 0.02), had less experience (8.45 vs. 9.79 years, *F* = 5.2; *p* = 0.02), and were more likely to have completed an undergraduate university degree (odds ratio (OR) 2.02, 95% confidence interval (CI) 1.44–2.83; *p* < 0.001), but less likely to practice at the advanced care paramedic certification (OR 0.61, 95% CI 0.42–0.88; *p* = 0.009). Our point estimates suggested women were also less likely to work full-time (OR 0.77, 95% CI 0.54–1.09) and more likely to be (or to have been) members of the peer support team (OR 1.25, 95% CI 0.59–2.65); however, neither difference reached the 5% significance threshold (*p* = 0.14 and 0.55, respectively).

### 3.3. Resilience

The internal consistency for the BRS in our survey was 0.85, consistent with previous investigations [45]. Across all participants, the average BRS score was 3.73 (95% confidence interval (CI) 3.68–3.79), corresponding to “normal” levels of resilience [53]. A total of 63 (10.6%) of our participants met the criteria for “low” levels of resilience. While mean BRS scores were higher among participants in a relationship (3.77 (SD 0.68) vs. 3.62 (SD 0.61), *p* = 0.002) and advanced (compared to primary) paramedics (3.82 (SD 0.65) vs. 3.69 (SD 0.67), *p* = 0.02), we did not observe any significant differences in the proportions of participants meeting the threshold for “low” resilience across demographic categories.

### 3.4. Mental Disorder Symptom Clusters

Internal consistency measures for the PCL-5 (α = 0.94), PHQ-9 (α = 0.87), and GAD-7 (α = 0.92) were all high and consistent with other investigations [55,60,61,62]. When stratified by subscale, internal consistency measures for Criteria B (α = 0.89), C (α = 0.83), D (α = 0.88), and E (α = 0.83) on the PCL-5 were all high. Mean reporting scores stratified by demographic characteristics are presented in Table 1. In total, 66 participants (11.2%) met the criteria for a positive screen for PTSD, 91 (15.4%) met the criteria for a positive screen for major depressive disorder, and 87 (14.7%) met the criteria for a positive screen for generalized anxiety disorder, with 145 participants (24.6%) screening positive for any of the three.

### 3.5. Unadjusted Models

In our unadjusted models (Table 2), age (OR 1.04, 95% CI 1.01–1.06 *p* < 0.001), experience (OR 1.05, 95% CI 1.02–10.7, *p* < 0.001), working full-time (OR 2.72, 95% CI 1.77–4.50, *p* = 0.003), being (or having been) a member of the peer support team (OR 3.03, 95% CI 1.42–6.45, *p* = 0.004), and “low” resilience (OR 8.15, 95% CI 4.62–14.36, *p* < 0.001) were all associated with an increased risk of screening positive for any one of PTSD, major depressive disorder, or generalized anxiety disorder.

The risk of a positive screen additionally varied across demographic categories, depending on the outcome being tested, the results of which are presented in Table 2.

### 3.6. Adjusted Models

Given the exploratory nature of our study, we included all demographic variables in our adjusted models. We also included interaction terms for gender × education (college versus university) and gender × employment status (part-time versus full-time) given the group differences we observed.

In our adjusted models, working full-time (OR 3.06, 95% CI 1.70–5.50, *p* < 0.001) and having “low” resilience (OR 10.41, 95% CI 5.59–19.40, *p* < 0.001) were the only characteristics associated with an increased risk of our composite outcome of a positive screen for any one of PTSD, major depressive disorder, or generalized anxiety disorder.

While “low” resilience persisted as a significant association when evaluating *mental disorder* symptom clusters, the associations with other demographic characteristics varied depending on the outcome being tested (Table 2).

## 4. Discussion

The goals of this study were to estimate the proportions of symptom clusters consistent with three specific *mental disorders* potentially associated with public safety work, and to explore the relationship between *mental disorder* symptoms and demographic variables and self-reported resilience. Because much of the extant research has relied on social media or email list servers to recruit participants, the concern is that the possibility of response bias may produce results that over or underestimate the true prevalence. In that respect, our response rate of 98% is a strength of our investigation, but our findings are simultaneously encouraging and concerning.

Among our sample, 11% of our participants screened positive for PTSD, 15% screened positive for major depressive disorder, and 15% screened positive for generalized anxiety disorder, with 25% screening positive for any one of the three *mental disorders*. Our estimates are lower than those reported among paramedics in a recent national study of *public safety personnel* in Canada [9]. This is encouraging because, while our findings are admittedly limited to a single site, it suggests that, when participant selection is carefully controlled, the prevalence of *mental disorder* symptoms among paramedics may be lower than has been previously described in this population. There are, however, some important caveats to our prevalence estimates that we discuss further in the limitations section.

Nevertheless, our findings are concerning for two reasons. First, our study supports the position that the prevalence of *mental disorder* symptoms among paramedics is significantly higher than rates observed in the general population in Canada [9,10,13,19,63]. In total, one in four of the active-duty paramedics in our study site met the screening threshold for any one of the three *mental disorders* we screened for—a problem that has likely only worsened since the emergence of the COVID-19 pandemic. What we do not know is the degree to which these participants have sought or are undergoing care for these symptoms. A number of studies have spoken to the stigmatization of *mental illness* within the public safety professions [64,65,66] and the reluctance of *public safety personnel* to seek professional help [2,66,67]. It is unfortunately likely that many are not receiving care at all. Understanding barriers to accessing mental health care among paramedics is an important topic for future research.

Our second objective was to examine the associations between various demographic characteristics and the risk of screening positive for PTSD, major depressive disorder, or generalized anxiety disorder. Previous research in the Canadian population would suggest that women are more likely than men to report current or past-year prevalence of any of the three, and that age, socioeconomic status, and education are also important predictors of *mental health* [68]. Similarly, recent findings point to differences in the risk of *mental disorder* symptoms among paramedics when stratified by gender, age, education, and relationship status [9,19,63]. Our findings both align and contrast with this body of research. First, we did observe differences in risk attributable to gender. In our adjusted models, women were more likely to screen positive for major depressive disorder and generalized anxiety disorder, but less likely for PTSD. Help-seeking behavior has been shown to differ across genders [69]. This plays out particularly with depression, where, among men, hegemonic conceptualizations of masculinity and stigma conspire to limit reporting and diagnosis [69,70]. Among our sample, it may be that women were more willing to disclose symptoms and—for PTSD—more likely to be off work because of this gendered difference in help-seeking. In total, 61 members of the paramedic service were on leave due to “disability” (using the language of the paramedic service) during the study, although we do not know the distribution of genders of the paramedics who were on leave. Further exploration of gendered differences in help-seeking and stigma in the context of public safety work is a topic worthy of further study.

Our findings also diverged from previous research among paramedics in the risk of *mental disorder* symptoms when stratified by relationship status, education, and provider classification. Whereas previous studies have found higher rates of *mental disorders* among advanced care paramedics [67] and protective effects of higher education [9,63] and being in a relationship [9,63], our findings did not bear this out. The only exception was where we found that participants with university-level education were half as likely to screen positive for generalized anxiety disorder. Given that women in our sample were more likely to attend university than men and more likely to screen positive for major depressive disorder or generalized anxiety disorder, we tested an interaction term between gender and education. While our point estimates for the term generally favored a protective effect (except for PTSD), our confidence intervals suggest that the effect is compatible with either an increased or a decreased risk of *mental disorder* symptoms. Where we did observe significant associations were in the relationships among self-reported resilience, employment classification, age, and experience as a member of the service’s peer support team. Of the three, resilience and experience as a peer supporter warrant careful consideration.

*Resilience* is generally held to be the degree to which an individual can “bounce back” from adversity [71]. The thought is that resiliency skills are teachable, and we have seen a growing trend of developing (and marketing) workplace resiliency training programs. When evaluated empirically, the effects of workplace resiliency training are modest [35,42,71,72,73,74], with research suggesting only small improvements in *mental disorder* symptoms, stigma reduction, or attitudes toward help-seeking. For example, a longitudinal study of Calgary police officers after completing the R2MR program did not indicate improvement in self-reported *mental disorder* symptoms or resilience at 6 or 12 months following the intervention [45]. As a whole, the topic of resiliency training is not without controversy. Although cultivating resilience may be desirable, the narrative risks shifting the locus of control onto the individual, potentially removing the responsibility of employers to mitigate the risks posed by chronic workplace *stressors* or exposure to *potentially psychologically traumatic events*. Controversy aside, our findings consistently and strongly point to a relationship between self-reported resilience and the risk of a positive *mental disorder* screen. The topic warrants further study.

Peer support as a concept has been the subject of discussion in the public safety professions for many years, owing, in part to, critical incident stress management programs [75]. More broadly, peer support was popularized by the consumer/survivor movement of the 1970s in which patients eschewed the (at the time) paternalistic medical models of psychiatric care in favor of seeking out the support of likeminded *people with lived experience* [76]. Among *public safety personnel*, evidence of peer support teams is mixed [77,78,79], but the health of the peer supporters themselves has not (to our knowledge) been studied. In our sample, we observed that being or having been a member of the service’s peer support team was associated with a fourfold increase in the risk of screening positive for PTSD and a more than threefold increase in the risk of screening positive for major depressive disorder. Interpreting this relationship is difficult. In our site, peer supporters were recruited on the basis of having lived experience with adversity, including *mental health challenges*, potentially confounding the association. It may be, for example, that paramedics who have struggled with mental health challenges are more likely to volunteer to help their peers. That said, recruitment for the program occurred 2 years before our study, and our self-report symptom measures probe for symptoms present within the last 14–30 days. Although it is certainly possible that members of the peer support team screened positive for *mental disorders* at higher rates due to persistent symptoms from pre-existing *mental health conditions*, it is also possible that the vicarious exposure to *potentially psychologically traumatic events* in providing empathetic support to their colleagues places peer supporters at an increased risk of poor *mental health*, including *mental disorders*. Given the growing popularity of these programs and the dearth of both effectiveness [78] and safety evidence, our findings emphasize the importance of studying peer support programs more closely, including the potential *health* consequences for peer supporters. This would require longitudinal studies with baseline health assessments of the peer supporters.

### Limitations

Our findings should be interpreted within the context of certain limitations. Firstly, cross-sectional research does not lend itself to establishing causality. We acknowledge that we are assuming that the *mental disorder* symptoms we studied are attributable in some way to the participants’ work as paramedics. This is increasingly supported by policy, however, given the growth in legislation in which a diagnosis of PTSD among *public safety personnel* is presumed to be work related to help facilitate access to treatment. Specifically for the relationship between resilience and the risk of a positive *mental* disorder screen, the directionality of the relationship is indeterminable. It may be, for example, that people who “have a hard time making it through stressful events” (a question on the Brief Resilience Scale) are indeed at an increased risk of mental illness because of this difficulty. Conversely, “feeling down, depressed, or hopeless”—a symptom in the Patient Health Questionnaire for major depressive disorder—may understandably make someone feel less resilient. Secondly, we acknowledge that self-report symptom measures, while widely used, are a surrogate outcome and not *diagnostic* in and of themselves. In that respect, our decisions to omit a specific Criterion A screen and to use the total symptom score in determining caseness also create an additional limitation. It is also worth mentioning that the participants may have been inclined to underreport symptoms given that the surveys were completed in a group setting and the potential feeling of being observed. Thirdly, while our study site was carefully selected to be illustrative of a large, sophisticated, urban paramedic service, it is nevertheless a single site, and readers must exercise caution in generalizing our findings. Our selected study site and participant recruitment methods were carefully chosen to strengthen the internal validity of the project, but necessarily traded off against external validity. Fourthly, our approach to modeling and subsequent statistical power depended on the event rate observed in our study. We could estimate this beforehand, but the basis for our study was predicated on an assumption of overestimated prevalence, the degree of which was difficult to know at the outset. We attempted to account for this in the design of our survey to limit the number of predictors, but we acknowledge a risk of overfitting our models. We would evaluate this risk as low, given that we had between seven and nine events per covariate [80]. Lastly, our study excluded 107 members of the service who were on leave during recruitment, making our sample vulnerable to a degree of selection bias, particularly given the 61 (8.5%) members who were on leave due to disability. The specific reasons for disability leave were unknown. Even so, we would suggest that our findings have unique value in reflecting the mental health of the active-duty workforce.

## 5. Conclusions

Our findings are encouraging in that the rates of symptom clusters consistent with various *mental disorders* that we observed in our study are lower than previously reported among paramedics in Canada. Although admittedly limited to a single site, this suggests that the ways in which we gather these data may have important implications for its interpretation. At the same time, the rates of *mental disorder* symptoms we observed are higher than reported in the Canadian population at large. One in four active-duty paramedics in our study met the screening criteria symptom clusters consistent with PTSD, major depressive disorder, or generalized anxiety disorder, pointing to a *mental health* crisis within the profession that—with the emergence of the COVID-19 pandemic—has likely only worsened. We echo the growing calls within scholarship and policy for urgent action to support the *mental health* and *wellbeing* of *public safety personnel* in Canada.

## Figures and Tables

**Figure 1 ijerph-19-04879-f001:**
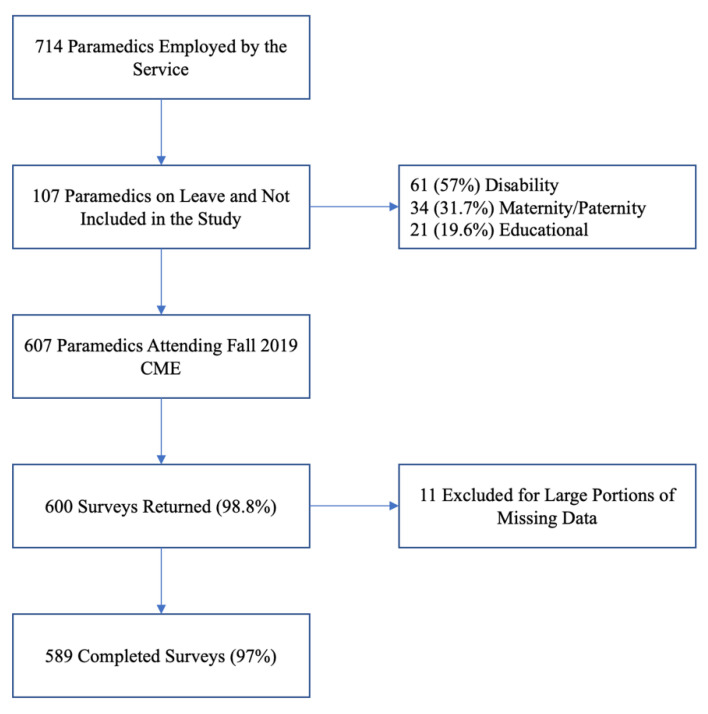
Participant flow diagram.

**Table 1 ijerph-19-04879-t001:** Mental disorder symptom scores (mean score) stratified by demographic category.

Demographic Category		*N* (%)	BRS	PCL-5	PHQ-9	GAD-7
			Range 1–5	Range 0–80	Range 0–27	Range 0–21
			Mean (SD)	Mean (SD)	Mean (SD)	Mean (SD)
All Participants		589	3.73 (0.67)	13.98 (4.16)	4.73 (4.74)	4.46 (4.76)
Gender	Men	354 (60%)	3.77 (0.65)	13.69 (0.72)	4.34 (4.28)	3.87 (4.23)
	Women	232 (39%)	3.67 (0.69)	14.50 (0.99)	5.34 (5.34) *	5.38 (5.38) ***
Relationship status	Single	143 (24%)	3.62 (0.61)	14.17 (14.28)	5.25 (4.79)	4.71 (4.75)
	Relationship	446 (76%)	3.77 (0.68) *	13.92 (14.14)	4.57 (4.72)	4.37 (4.70)
Employment	Part-time	194 (33%)	3.74 (0.66)	10.85 (11.90)	3.70 (4.32)	3.29 (4.08)
	Full-time	394 67%)	3.73 (0.67)	15.56 (14.92) ***	5.24 (4.86) ***	5.04 (4.97) ***
Education	College	330 (56%)	3.71 (0.67)	14.77 (14.33)	4.97 (4.97)	4.75 (4.84)
	University	259 (44%)	3.76 (0.65)	12.98 (13.91)	4.42 (4.60)	4.09 (4.64)
Provider classification	PCP	398 (67%)	3.69 (0.67)	13.30 (13.62)	4.77 (4.82)	4.45 (4.82)
	ACP	188 (32%)	3.82 (0.65) *	15.42 (15.16)	4.64 (4.57)	4.48 (4.64)
Peer support team	Member	29 (45%)	3.54 (0.82)	23.41 (22.59) ***	7.76 (6.42) ***	7.38 (6.50) **
	Nonmember	560 (95%)	3.74 (0.66)	13.53 (13.46)	4.58 (4.60)	4.31 (4.62)

SD = standard deviation; BRS = Brief Resilience Scale; PCL-5 = Post-Traumatic Stress Disorder Checklist; PHQ-9 = Patient Health Questionnaire; GAD-7 = Generalized Anxiety Disorder. * *p* < 0.05 ** *p* < 0.01 *** *p* < 0.001.

**Table 2 ijerph-19-04879-t002:** Positive mental disorder screens stratified by demographics and resilience in unadjusted analyses and positive mental disorder screens stratified by demographics and resilience in adjusted analyses.

All Participants (*N* (%))	BRS < 3.00		PCL-5 > 32		PHQ-9 > 9		GAD-7 > 9		Any	
	63 (10.7%)		66 (11.2%)		91 (15.4%)		87 (14.7%)		145 (24.6%)	
Covariate	OR	95% CI	OR	95% CI	OR	95% CI	OR	95% CI	OR	95% CI
Age	0.99	0.96–1.02	1.06 ***	1.02–1.09	1.04 ***	1.02–1.07	1.03 *	1.00–1.06	1.04 ***	1.01–1.06
Experience	0.99	0.95–1.02	1.05 ***	1.02–1.09	1.04 **	1.01–1.07	1.03 **	1.00–1.06	1.05 ***	1.02–1.07
Women	1.35	0.79–2.27	0.73	0.43–1.26	1.61 *	1.02–2.52	1.88 **	1.19–2.98	1.38	0.94–2.02
Single	1.38	0.78–2.46	0.81	0.43–1.52	1.21	0.73–2.00	1.13	0.67–1.90	1.25	0.81–1.91
Full-time	0.78	0.45–1.34	2.72 **	1.39–5.33	2.87 ***	1.60–5.16	2.63 ***	1.50–4.83	2.82 ***	1.77–4.50
College	0.97	0.57–1.64	1.53	0.89–2.61	1.32	0.78–1.94	1.78 *	1.10–2.89	1.43	0.97–2.11
ACP	0.76	0.42–1.36	1.54	0.91–2.60	0.98	0.61–1.59	1.00	0.61–1.63	1.06	0.71–1.58
Peer supporter	1.35	0.45–4.01	4.70 ***	2.08–10.62	4.72 ***	1.96–9.28	2.76 *	1.21–6.28	3.03 **	1.42–6.45
“Low” resilience (BRS < 3.00)			9.30 ***	5.12–16.89	6.34 ***	3.61–11.12	6.29 ***	3.57–11.08	8.15 ***	4.62–14.36
**Model ((χ^2^ (*p*))**			**PCL-5 > 32**		**PHQ-9 > 9**		**GAD-7 > 9**		**Any Positive Screen**	
			**82.19 (*p* < 0.001)**		**78.30 (*p* < 0.001)**		**72.14 (*p* < 0.001)**		**101.3 (*p* < 0.001)**	
**Covariate**			**OR**	**95% CI**	**OR**	**95% CI**	**OR**	**95% CI**	**OR**	**95% CI**
Age			1.07 *	1.00–1.13	1.06 *	1.00–1.11	1.02	0.96–1.08	1.02	0.97–1.07
Experience			0.96	0.90–1.03	0.97	0.91–1.03	0.99	0.93–1.05	1.00	0.95–1.06
Women			0.73	0.39–1.37	1.78 *	1.07–2.95	2.20 **	1.32–3.67	1.52	0.98–2.34
Single			0.95	0.46–1.96	1.35	0.76–2.40	1.22	0.68–2.19	1.45	0.88–2.34
Full-time			2.15	0.93–4.96	2.85 **	1.41–5.77	2.85 **	1.41–5.77	3.06 ***	1.70–5.50
University			0.70	0.37–1.32	0.82	0.48–1.39	0.50 *	0.29–0.86	0.70	0.45–1.10
Advanced care			1.19	0.62–2.28	0.77	0.40–1.29	0.82	0.45–1.46	0.76	0.46–1.24
Peer supporter			4.05 **	1.57–10.43	3.31 **	1.40–7.84	2.07	0.83–5.14	2.13	0.92–4.89
“Low” resilience (BRS < 3.00)			13.09 ***	6.70–25.54	7.65 ***	4.14–14.10	7.33 ***	3.96–13.55	10.41 ***	5.59–19.40
**Model (χ^2^ (*p*))**			**0.41 (*p* = 0.81)**		**0.911 (*p* = 0.63)**		**0.75 (*p* = 0.68)**		**1.06 (*p* = 0.58)**	
**Interaction Term**			**OR**	**95% CI**	**OR**	**95% CI**	**OR**	**95% CI**	**OR**	**95% CI**
Gender × university			1.48	0.42–5.32	0.62	0.22–1.75	0.97	0.33–2.85	0.68	0.28–1.66
Full-time × gender			0.90	0.19–4.19	0.77	0.22–2.72	0.56	0.15–2.07	0.70	0.24–1.97

SD = standard deviation; OR = odds ratio; PCP = primary care paramedic; ACP = advanced care paramedic; BRS = Brief Resilience Scale; PCL = Post-Traumatic Stress Disorder Checklist; GAD = Generalized Anxiety Disorder. “Any” = any positive screen excluding BRS. * *p* < 0.05; ** *p* < 0.01; *** *p* < 0.001.

## Data Availability

The data presented in this study are available on request from the corresponding author. The data are not publicly available due to privacy restrictions and data security procedures stipulated in the Research Ethics Board (REB) review of this project.

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
