# Peer review of "Mental Disorder Symptoms and the Relationship with Resilience among Paramedics in a Single Canadian Site"

_ijerph, 2022, doi:10.3390/ijerph19084879_

Round 1

Reviewer 1 Report

Thank you for the opportunity to review the paper “Mental disorder symptoms among paramedics in a single Canadian site”.

I found the paper interesting, highly relevant, well-written, clear, and well-discussed. Accordingly, I have only two minor comments which should be added to the discussion / limitation section:

As a further limitation with respect to the cross-sectional design, it should be explicitly added that the relationship between resiliency and the different outcome measures could be the other way around or bi-directional.

Similarly, membership in the peer support system should be further discussed: while authors argue that the recruitment was probably biased, also the agreement of participation (after being asked) could be biased: it might be that paramedics who suffer themselves are more likely to engage in helping others. Long-term studies are needed to explore the causal relationships here, with baseline measures when paramedics initially join the peer support system.

Author Response

Dear Colleague,

Thank you for your very helpful review of our paper. We have made both edits you suggested. Pease see the attachment for a more detailed explanation.

Many thanks,

Justin

Reviewer 2 Report

The article "Mental disorder symptoms among paramedics in a single Canadian site" aims to estimate the proportion of mental disorder symptoms among paramedics in a sample from Ontario (Canada) and its relationship with resilience.

The manuscript is well written and has been pleasant to read. Material and methods are amply described, giving an appreciation of the detail with which the study has been designed. 

However, I have some comments listed below:

-Despite careful control of participant selection and survey procedure, participants may have underreported their symptoms due to the Hawthorne effect, feeling over-observed and over-controlled. This underreporting is also supported by the reluctance of SPS to talk about their symptoms due to stigma, as you noted in lines 341-343. This should be included as a limitation. In addition, I would like to know if it was taken into account whether participants had enough time to complete the survey comfortably and without rushing.

-82,5% of the paramedics employed by the service have participated in the study. However, there is an 8,5% that were not included due to disability. The reasons for disability are not specified, and some of them could be due to psychological reasons, underestimating the prevalences given by the study.

-Line 258: The authors pointed out an OR of 1.05 in “experience”, but in the table 2A they have indicated an OR of 1.04.

-Line 260: The authors pointed out an OR of 3.03 in “member of the peer support team”, but in the table 2A they have indicated an OR of 3.02.

-Line 331-333: This statement is difficult to confirm because of the possible Hawthorne effect and prevalence underestimation due to the absence of personnel with disability. Furthermore, although the internal validity of the study is high, its external validity remains to be determined due to the geographic specificity of the study.

-Line 393-394: Despite of the controversy between resilience and mental disorder symptoms appearence, further explanation of the possible causes in the study sample would be advisable.

Author Response

Dear Colleague,

Thank you for your very helpful review of our paper. We have made all of the edits you suggested. Please see the attachment for a more detailed explanation.

Many thanks,

Justin

Reviewer 3 Report

I find the research problem important and interesting.

The paper, in order to be accepted, needs a complete rewrite. In reality, it should be rewritten from the ground up.

The title: It should be reformulated. Many different variables have been taken into consideration in the research.

Introduction. Actually there is no introduction. Information about the context of the problem is important but should be only a short introduction at the beginning of the theoretical part. Very important problems like, PTSD, Resilience, anxiety disorder, depressive disorder are not described at least succinctly. No justification was given as to why these issues were chosen. In short, no theoretical basis for the research. No short reference to research that has already been done on this topic among health professionals. Similar studies are plentiful.

Lack of clearly formulated research problem. No hypotheses. Only a laconic statement in lines 155-156: "that self-reported resilience scores will have an inverse relationship with the risk of testing positive for mental disorders".  A lot of data are described but no hypotheses are formulated for them. These should have been formulated in the introduction. Collected together or formulated sequentially with theoretical justification. This is a serious methodological error of the article. No information about what were the dependent variable and independent variables.

Research tools. There is information about them in many places (Methods and Results). This needs to be sorted out.  E.g. the PTSD Checklist-5 (PCL-5) consists of five subscales, and only the value of Cronbach's alpha coefficient for the overall scale is presented. When describing the tool, information about its psychometric properties should be collected.

A similar comment applies to respondents. Data scattered in several places. Divergent information on number of respondents. Presentation of data should be reviewed and sorted. Line 208-210 - 589 participants; Line 220-360 "In total 354 our participants"). N data in the description of some variables (Table 1) do not agree. The inclusion of Figure 1 does not make sense. Lack of criteria, inclusion and exclusion in the study group.

Results

The use of the (weak test) non-parametric 'chi-square tests for continuous and categorical data' for the analysis is questionable. Do the distributions of continuous variables not satisfy the normal distribution condition? This should have been preceded by the Kolmogorov-Smirnov test or the Lilliefors test for Normality & Exponential Distributions. A better methodological solution would have been to use advanced analyses e.g. multiple regression with variable entry or stepwise technique. The predictors would be rank ordered as to their impact.

Very general comment on the large amount of information contained in Tables 1, 2a and 2b.

Discussion. The best written piece of text, although too vague

References: Contemporary literature

Author Response

Dear Colleague,

Thank you for your very helpful review of our paper. Please see the attachment for a detailed response to your comments.

Many thanks,

Justin

Round 2

Reviewer 2 Report

Although the study is generally well written, as I have pointed out there are some methodological flaws that may have incurred a bias, underestimating the prevalences given. I recognize the authors' interest in including these biases in the limitations section, but I consider that these weaknesses cast doubt on the results shown, compromising the aim of the study.

For these reasons, I must unfortunately reject the article. I wish the authors the best of luck.

Author Response

Dear Colleague,

I must admit to being at something of a loss for how to respond to your latest feedback. 

In your first review of our paper, you very kindly described our manuscript as “well written and has been pleasant to read”, going on to say that “Material and methods are amply described, giving an appreciation of the detail with which the study has been designed.” On the scoring rubric, you also indicated that the research design was appropriate and that the methods were adequately described (i.e., “Yes” in both categories). You made recommendations that were specific, constructive, very insightful and which - in my view - strengthened the quality of our manuscript. I was quite taken aback to see that after making the changes you recommended, the score for the appropriateness of the research design had suffered and that you were recommending rejection.

Neither my team nor I have encountered a situation where a reviewer, in the first instance, recommends minor revisions only to then reject the paper when the recommended edits are made. It is admittedly puzzling to find myself in a situation where the design of the study between reviews has not changed, but a reviewer’s evaluation of it has so drastically. Being unsure how to proceed, I will defer to the judgment of the decision editor on submission of this latest revision.

In the meantime, however, let me speak in defense of our research design in the hopes that I can convey to you that our approach was indeed very carefully thought out.

In choosing to distribute our survey during an in-person continuing medical education session, we drew on similar research conducted among paramedics in Ontario that used the same approach. For example, Bigham and colleagues (2014) used exactly this method when surveying paramedics from Ontario and Nova Scotia on their experiences with workplace violence - a survey that (like our own) included personal and potentially sensitive questions. Similar work by Halpern and colleagues (2011, 2012) recruited paramedics in Toronto, Ontario during CME sessions to fill out cross-sectional surveys on critical incident stress. In practice, paramedics in our context are rather accustomed to being surveyed during CME sessions - up until the recent events with the COVID-19 pandemic, the paramedics would commonly complete employee satisfaction surveys during CME as well as questionnaires on a variety of topics relevant to the service. Alongside our survey package, for example, the paramedics also completed a questionnaire on their preferences for online CME content. 

Nevertheless, we recognized that the questions included in our survey were indeed quite sensitive and we wanted to ensure that the paramedics would feel as comfortable as could reasonably be expected in participating. Our research team brings a multidisciplinary perspective to mental health, with members having expertise in cognitive psychology, clinical social work, occupational therapy, and psychiatry, including familiarity with the paramedic context. In addition to a robust ethics review of our procedures, we also consulted with the paramedic service’s leadership team, psychological health and safety committee, joint health and safety committee, and the elected leadership of the union representing the paramedics in the service. We delivered messaging ahead of the CME to inform service members that they would be invited to participate in the survey during their CME and that such participation was entirely voluntary. This was in addition to an in-person presentation I delivered on the morning of each CME day. Members who did not want to complete the survey were told they could leave the package blank (or fill the pages with doodling) and place the survey in the included opaque envelope and deposit it in the drop box with the rest of the class - an effort to ‘disguise’ non-participation. They were free to keep the gift card with our thanks. Only a few did, and a surprising number returned their gift cards to me, insisting that I donate them to people in need (i.e., our community’s underhoused people). 

To the last point on the exclusion of 61 members (8% of service) on leave due to disability, it is regrettable that we were not able to survey the members. Things changed quickly in March 2020 (when we planned to mail the surveys), and even I was redeployed to help coordinate our service’s response to the crisis. Having said that, what strikes me most about our findings - and I think contributes to the urgency of publishing - is that 25% of the active-duty paramedic workforce met the screening criteria for at least one mental disorder. In my view this lends credible evidence to the existence of a mental health crisis within the profession that, in the months since the pandemic began, has likely only worsened.

Regardless of the ultimate decision on our manuscript, please accept our gratitude for your time in reviewing the paper.

Warm regards,

Justin

Reviewer 3 Report

I have read the authors' responses and the revised version of the manuscript. Several of my comments were taken into account by the authors, which improved the quality of the paper. In other cases, they shared their comments. They are entitled to do so. 
However, for the sake of the high quality of the study and the accurate description of it, I still expect a correction of three points.
(1). Title. I understand that this is a challenge for the authors (they described it in the commentary). However, I cannot agree that resilience is a symptom of mental disorder. It is a dispositional variable, possibly a process (resiliency). Resilience was also included by the authors in the hypothesis. So why describe this variable if it is left out of the title? The authors have to decide on something. If they do not drop resilience, the title should be clarified and the introduction completed (2).
(3) Since the International Journal of Environmental Research and Public Health is the best scientific interdisciplinary journal with a global scope targeting a large audiences, all research instruments used in the study should include their psychometric properties (3). The use of even the "most perfect" instrument does not absolve researchers from this obligation.
If the authors expect my positive recommendation to publish the manuscript, completing these three points is essential.  

When interpreting the results of skewness and kurtosis: "Although both tests were significant at the 5% level for all continuous variables (indicating non-normal distributions), neither test exceeded a value of 2 or 9 for skewness or kurtosis, respectively, and thus should not violate the assumption of normality in the parametric tests we use significantly." I understand that a "2-9!!!" error has crept in here. (George, D., & Mallery, P. (2016). IBM SPSS Statistics 23 Step by Step: A Simple Guide and Reference (14th ed.). New York: Routledge. https://doi.org/10.4324/9781315545899)

Author Response

Dear Colleague,

Thank you again for taking the time to review our manuscript and provide feedback on our work. Please see the attachment for a more detailed response to your comments.

Many thanks,

Justin
